# Polyphenol Extracts from Three Colombian Passifloras (Passion Fruits) Prevent Inflammation-Induced Barrier Dysfunction of Caco-2 Cells

**DOI:** 10.3390/molecules24244614

**Published:** 2019-12-17

**Authors:** Juan Carlos Carmona-Hernandez, Gonzalo Taborda-Ocampo, Jonathan C. Valdez, Bradley W. Bolling, Clara Helena González-Correa

**Affiliations:** 1Grupo de Investigación Médica, Universidad de Manizales, Manizales 170004, Colombia; 2Grupo de Investigación Cromatografía y Técnicas Afines, Universidad de Caldas, Manizales 170004, Colombia; gtaborda@ucaldas.edu.co; 3Department of Food Science, University of Wisconsin-Madison, Madison, WI 53706, USA; jcvaldez@wisc.edu (J.C.V.); bwbolling@wisc.edu (B.W.B.); 4Grupo de Investigación Nutrición, Metabolismo y Seguridad Alimentaria, Universidad de Caldas, Manizales 170004, Colombia; clara.gonzalez@ucaldas.edu.co

**Keywords:** Colombian *Passifloras*, intestinal barrier function, transepithelial electrical resistance, Caco-2 cells, flavonoids

## Abstract

Chronic intestinal inflammation is associated with pathophysiology of obesity and inflammatory bowel diseases. Gastrointestinal inflammation increases barrier dysfunction exacerbating the immune response and perpetuating chronic inflammation. Anti-inflammatory flavonoids may prevent this intestinal barrier dysfunction. The purpose of this study was to evaluate the polyphenol composition of Colombian *Passiflora*
*edulis* var. Flavicarpa (Maracuyá), *Passiflora*
*edulis* var. Sims (Gulupa), and *Passiflora*
*ligularis* var. Juss (Granadilla) (passion fruits) and to evaluate their ability to inhibit disruption of intestinal barrier dysfunction of Caco-2 (colorectal adenocarcinoma) cells by an inflammatory cocktail (IC). Polyphenols (flavan-3-ols, phenolic acids, flavonols), xanthenes, and a terpene were identified in passion fruits. Cyanidin 3-rutinoside, (+)-catechin and ferulic acid were the most abundant phenolics in *P*. *edulis* var. Flavicarpa, *P.*
*edulis* var. Sims, and *P.*
*ligularis* var. Juss, respectively. Fruit extracts prevented loss of transepithelial electrical resistance in Caco-2 cells treated with the IC. Among the extracts, *P. ligularis* var. Juss was most effective at maintaining Caco-2 transepithelial electrical resistance (TEER) with ~73% relative to the IC-treated cells with about 43% of initial TEER values. This fruit had cyanidin-3-rutinoside, (+)-catechin, (−)-epicatechin, and ferulic acid in its phenolic profile. Results of this work support the hypothesis that consumption of passion fruit extracts could benefit intestinal health.

## 1. Introduction

Chronic inflammation is a hallmark of many non-communicable diseases [1]. Dietary approaches to prevent or mitigate chronic inflammation are needed to reduce disease risk [2]. Recently, intestinal barrier homeostasis has been recognized as an important contributor to development of chronic inflammation [2]. Chronic inflammation leads to a dysfunctional intestinal barrier, and hinders the resolution of inflammation. Thus, inflammation-induced barrier dysfunction is associated with obesity, inflammatory bowel disease, cardiovascular disease, and colon cancer [2,3].

Polyphenols are promising dietary anti-inflammatory compounds [2,4]. Consumption of polyphenols inhibits chronic intestinal inflammation in mice [4]. Furthermore, increased polyphenol consumption can also inhibit colon cancer in mice [5]. Polyphenol content of fruits varies due to climate, species, maturation status, and soil condition among others. Generating country-specific information related to tropical fruit polyphenols is useful in determining their potential to reduce chronic disease risk. The biodiversity of Colombian fruits and vegetables could be useful as nutritional sources of polyphenol-rich foods. Colombian nutritional databanks do not currently include polyphenols and flavonoid content of regional foods and their health benefits. Thus, it is necessary to determine polyphenols in Colombian fruits and vegetables to ensure the long-term health of local communities and to improve their agricultural economy.

There is worldwide interest in increasing the production of Colombian fruits and vegetables. Colombia is the ninth supplier of exotic fruits worldwide, and has increased from 233 million to 1.2 billion consumers in less than 10 years [6]. Specifically, uchuva, tree tomato, tamarind, and granadilla (passion fruit) demand is increasing. Polyphenols and flavonoids found in fruits from the Andean and Amazonic regions, could offer nutritional and health benefits. Several types of the *Passiflora* species are also harvested in these regions; some examples are *Passiflora edulis* var. Flavicarpa, yellow passion fruit (Maracuyá), *Passiflora edulis* var. Sims, purple passion fruit (Gulupa), and *Passiflora ligularis* var. Juss, sweet granadilla (Granadilla), fruits represented in Figure 1.

Despite multiple studies highlighting the uses of tropical fruit flavonoids for the control of free radicals (natural antioxidants) [7,8], or the regulation of fungal and bacterial activities [9], information on anti-inflammatory activity is lacking. Granadilla, Gulupa, and Maracuyá (passion fruits) are promising fruits for further study because of their proven medicinal benefits [8]. The present work aims to identify and quantify polyphenols extracted from *P. edulis* var. Flavicarpa (Maracuyá), *P. edulis* var. Sims (Gulupa), and *P. ligularis* var. Juss (Granadilla), gathered in the Colombian coffee region, and to evaluate the possible in vitro inhibitory action on inflammation-disrupted intestinal barrier function in Caco-2 cells.

## 2. Results

### 2.1. UHPLC-MS Analysis of Passiflora Extracts

Colombian *Passiflora* fruit polyphenols were identified by ultra-high performance liquid chromatography tandem mass spectrometry (UHPLC-MS) in electrospray ionization mode through comparison to reference standards. Subsequently, targeted quantification was applied to characterize the phytochemical profiles of extracts from *P. edulis* var. Flavicarpa (Maracuyá), *P. edulis* var. Sims (Gulupa), and *P. ligularis* var. Juss (Granadilla). A total of 16 compounds were detected that included polyphenols (flavan-3-ols, anthocyanins, phenolic acids), xanthenes, and a triterpenoid were identified in each of the three passiflora samples (Table 1). Example chromatograms for reference standards are presented in Figure A1 (Appendix A).

The concentrations of phytochemicals in *Passifloras* fruit extracts were determined for compounds detected above the limit of quantification (LOQ) (0.05 µg/mL) (Table 2). The main components detected in 70% methanol in water (*v*/*v*) extracts of passion fruits were phenolic acids, xanthines, catechins, and anthocyanins. (+)-Catechin, (−)-epicatechin, ferulic acid, cyanidin 3-rutinoside, and quercetin 3-glucoside were the main polyphenols.

Representative chromatograms in Figure 2 illustrate peaks, retention times, and *m*/*z* values for cyanidin 3-rutinoside and ferulic acid, the polyphenols with the highest concentration, for *P. edulis* var. Flavicarpa (Maracuyá). In contrast, caffeine, cyanidin, pelargonidin, and luteolin were below the limit of quantitation in yellow passion fruit (Maracuyá).

Phenolic acids and catechins were detected in *P. edulis* var. Sims (Gulupa) (purple passion fruit) (Figure 3). The most abundant polyphenols detected in purple passion fruit were (+)-catechin, (−)-epicatechin, and rosmarinic acid. However, caffeine, quercetin 3-glucoside, cyanidin and pelargonidin were less than the LOQ in purple passion fruit extracts.

The predominant polyphenols detected in Granadilla extracts were ferulic acid, epigallocatechin, epigallocatechin gallate, and (−)-epicatechin (Figure 4). Compounds, in this *Passiflora*, detected less than LOD were (+)-catechin, rosmarinic acid, and cyanidin 3-rutinoside. The last two compounds were quantified in purple and yellow passiflora at 0.13 and 0.60 µg/mL, respectively.

The yellow passion fruit extract yielded, quantifiable amounts of cyanidin 3-rutinoside, ferulic acid, (+)-catechin, (−)-epicatechin, theobromine, and quercetin 3-glucoside. Comparing it to purple passion fruit, (+)-catechin, and (−)-epicatechin were also quantifiable as well as rosmarinic acid. With respect to sweet granadilla, ferulic acid, (−)-epicatechin, epigallocatechin gallate, and epigallocatechin were detected above LOD. Total concentration of quantifiable compounds detected in the extracts are presented in Table 3; these concentrations are contrasted with total content per dry weight (DW) and total polyphenol content (TPC) in the 70% methanol extracts.

### 2.2. Effect of Passion Fruit on Caco-2 Barrier Function

Caco-2 cells were exposed to pro-inflammatory cytokines and LPS to induce barrier dysfunction. The inflammatory cocktail (IC) reduced barrier function, determined by transepithelial electrical resistance (TEER), by approximately half of the initial value at 36 h (Table 4). Over the same time course, Caco-2 cells increased TEER by 9%. Cells, without exposure to IC, maintained barrier function for up to 48 h.

Quercetin was applied to Caco-2 cells in the presence of IC as a flavonoid reference standard (Figure 5). At 5 mg/mL, quercetin did not inhibit barrier dysfunction induced by the IC, and declined in a similar manner to the IC control over 60 h.

Next, methanolic extracts of passifloras, and the residual products of the extractions, were applied to Caco-2 cells in the presence of IC. Different concentrations were tested looking the monitor the % TEER with respect to the response on barrier integrity. Figure 6 shows a comparative decreasing TEER percentage for polyphenol extracts, at 10.0 mg/mL, from Colombian Granadilla, Gulupa, and Maracuyá acting on IC. At 60 h, the IC had reduced Caco-2 TEER to 41.7% of the initial values. Here, Caco-2 cells treated with Gulupa, Maracuyá, and Granadilla had TEER values of 60.8, 63.3, and 72.7% of initial values, respectively.

The dose responses of methanolic extracts of passifloras were tested for inhibition of IC-induced barrier dysfunction in Caco-2 cells (Table 5). At 48 h, Granadilla inhibited Caco-2 barrier dyfunction more effectively than Gulupa and Maracuyá at 10 mg/dL. The extraction residue Gulupa was more effective at inhibiting Caco-2 barrier dysfunction than its methanolic extract at 10 mg/dL.

Although there where statistical significant differences at 36, 48, and 60 h for the treatments compared in Table 4, the 48 h point yielded the best results in TEER for the Caco-2 cells exposed to the IC and the residues of the extracts from the three Colombian *Passifloras*.

## 3. Discussion

In this study, we characterized the phytochemicals present in three Colombian passion fruits (*Passifloras*) and evaluated the ability of their extracts to inhibit inflammation-induced Caco-2 barrier dysfunction. A limited number of studies have investigated the composition of fruits from passifloras. In two prior studies, pelargonidin, quercetin, and luteolin derivatives were identified in *P.* edulis f. Flavicarpa, *P. edulis, P. alata*, and *Passiflora ligularis* [7,8]. However, in this study, cyanidin 3-rutinoside, ferulic acid, (+)-catechin, and (−)-epicatechin were identified in Colombian passion fruits. In purple passion fruit, (+)-catechin, (−)-epicatechin and rosmarinic acid were identified as the most abundant compounds. A similar study evaluated polyphenol content in this Colombian fruit [9], finding quantifiable concentration of caffeic, *p*-coumaric and ferulic acids, with no detection of (+)-catechin, and (−)-epicatechin, cyanidin 3-rutinoside, and quercetin 3-glucoside. *Passifloras* from Brazil and India (*P. edulis* f. Flavicarpa and *P. subpeltata*, synonym *P. calcarata*) also have similar polyphenol profiles to those reported in the present study [8,10]. However, none of the previous reports have identified ferulic acid as the most abundant polyphenol, as such with *P. ligularis* var. Juss (Granadilla) in the present study. Epigallocatechin, epigallocatechin gallate, and (−)-epicatechin were the other main polyphenols of Colombian passion fruits in the present study. Other less abundant compounds were caffeine, epicatechin, luteolin, apigenin, and quercetin 3-glucoside, which were also reported in other studies of *Passifloras* fruits [8,10].

Colombian passifloras inhibited inflammation-induced intestinal barrier dysfunction in Caco-2 cells. TEER is regarded as a reliable indicator of the normal and functional state of cell membranes analysis and has been applied to a variety of cell types [11]. The integrity of the intestinal barrier is maintained by tight junction (TJ) proteins [12]. Maintenance of these cellular structures are essential for the protection of bacterial translocation and leakage of pro-inflammatory compounds from the gut [13,14]. A dysfunctional intestinal barrier is associated with obesity and inflammatory bowel diseases [15].

Maintaining intestinal barrier homeostasis is important for digestive health [16]. Caco-2 cell models originated from human adenocarcinoma, have been commonly used to evaluate potential inhibitors of inflammation-induced intestinal barrier dysfunction [17,18,19]. In Caco-2 cells, pro-inflammatory interleukins, such as TNF-α and IL-1β, activate cell signaling pathways that affect TJ and membrane integrity [20]. As a result, inflammatory stimuli reduce TEER values as a consequence of disrupted monolayers due to the loss of TJ integrity [20].

Crude polyphenols extracts and yogurt have previously been reported to improve TEER values in different in vitro and in vivo models of barrier function [4,16,21,22,23]. The improvement in TEER results has been demonstrated by the inhibitory action of different polyphenols on inflammatory routes related to cyclooxygenases (COX) type COX-1 and COX-2, nuclear factor kappa B (NF-κB), and by the increasing tight junction proteins ZO-1 and occludin [4]. In the present study, polyphenols extracted for Colombian *P. edulis* var. Flavicarpa (Maracuyá), *P. edulis* var. Sims (Gulupa), and *P. ligularis* var. Juss (Granadilla) improved TEER values on Caco-2 cells stimulated treated with IC. However, further work is needed to investigate the mechanism(s) of action for improving TEER.

Positive TEER results from *Passiflora* extracts were dose-dependent. Lower doses of 2.5 and 5.0 mg/mL did not improve Caco-2 barrier dysfunction. However, at 10 mg/mL, equivalent to approximately 10–25 mM polyphenols, improved Caco-2 barrier dysfunction. Notably, quercetin alone at 5 mg/mL was not sufficient to inhibit IC-induced TEER dysfunction in Caco-2 cells. Thus, a diverse polyphenol profile appears more favorable to inhibit intestinal barrier dysfunction in Caco-2 cells.

Colombian passion fruits had varying polyphenol profiles, which may explain differences in bioactivity between extracts. Granadilla, the most active fruit extract 10 mg/mL, was enriched in epigallocatechin and ferulic acid. In comparison, Maracuyá was enriched in cyanidin-3-rutinoside, ferulic acid, (+)-catechin and (−) epicatechin, whereas Gulupa had (+)-catechin, (−)-epicatechin, and rosmarinic acid. Considering compound quantification and synergistic effect among them, the suggested compound combination more responsible for the TEER effects would be in Granadilla ferulic acid (high content) with (−)-epicatechin (low content), in Gulupa ferulic acid (low content) with (+)-catechin and (−)-epicatechin (midlevel content), and in Maracuyá ferulic acid (high content) with (+)-catechin and (−)-epicatechin (midlevel content). The compound levels, for this comparative aspect, are based on 0.05−0.15 µg/mL as low contents; 0.16−0.30 µg/mL as midlevel contents; and 0.31−0.60 µg/mL as high contents. Thus, fruits having ferulic acid with epigallocatechins appeared to be more effective inhibiting IC-induced barrier dysfunction in Caco-2 cells. Prior studies have also demonstrated the ability of FA to inhibit inflammation and protect gastrointestinal cells and TJ [24,25,26]. FA in Gulupa, Maracuyá, and Granadilla extracts showed TEER improvement (Table 3) of Caco-2 cells.

Although polyphenols exhibited direct effects in the present study, it is possible that instability during digestion or the host and microbial metabolism would change the polyphenol profile in the gut. In example, anthocyanins are extensively catabolized to phenolic acids by gut microbiota [27] Likewise, (−)-epicatechin is catabolized to dihydroxyphenyl-γ-valerolactones in the intestine [28]. It is important to consider the stability of anthocyanins and other phenolics during digestion. In a human pharmacokinetic study, a significant proportion of 13C cyanidin-3-glucoside was detected as microbial catabolites in the feces [29]. In cell culture media, phenolics have varying stability [30]. Anthocyanins are generally stable during model digestion through the stomach, but less stable during the intestinal phase and form varying phenolic acid products [31]. Grape skins subjected to in vitro digestion retained 40−80% of the polyphenols [32]. Thus, in vivo studies of passion fruits are needed to confirm the activity demonstrated by the present study.

In conclusion, Colombian passion fruits have diverse phytochemical profiles and are dietary sources of polyphenols. Given the ability of passion fruit extracts to inhibit IC-induced barrier dysfunction in Caco-2 cells, further work on elucidating mechanisms of action are warranted. This study suggests that consumption of Colombian passion fruits could be relevant to diseases involving a compromised intestinal barrier, including obesity, inflammatory bowel diseases, diabetes, and colon cancer.

## 4. Materials and Methods

### 4.1. Standard Reference Solutions

Caffeine, theobromine, teofiline, (±)-catechin, (−)-epigalocathechin gallate, (−)-epicatechin, (−)-epicathechin gallate, (−)-epigallocatechin, cafeic acid, *p*-coumaric acid, vanillic acid, rosmarinic acid, quercetin, naringenin, luteolin, kaempferol, ursolic acid, pinocembrine, carnosic acid, apigenin, ferulic acid, cyanidin-3-rutinosede, pelargonidin-3-glucoside, cyanidin, pelargonidin, quercetine 3-glucoside, and kaempferol 3-glucoside were from Sigma Aldrich (St. Louis, MO, USA). Dulbecco’s Modified Eagle Medium (DMEM, high glucose), qualified fetal bovine serum (FBS), MEM non-essential amino acid solution (NEAA), Hank’s balanced salt solution (HBSS), penicillin/streptomycin, and TRIzol reagent were purchased from Life Technologies (Grand Island, NY, USA). Interleukin 1 beta (IL-1β), tumor necrosis factor alpha (TNF-α), interferon gamma (IFN-γ), lipopolysaccharide (LPS), bovine serum albumin (BSA), and trypsin were obtained from Sigma Aldrich (St Louis, MO, USA). Transwell plates and inserts (24-well plates) were acquired from Corning, Inc. (Lowell, MA, USA).

### 4.2. Sample Preparation

All laboratory procedures and fruit sampling were approved by the Bioethical Committee from Universidad de Caldas (Document CBCS-036, Acta No. 012 de 2017). Samples of *Passifloras* were collected from the Colombian States of Caldas, Tolima, and Valle del Cauca. Granadilla was collected in Aranzazu (Caldas), Gulupa in Cajamarca (Tolima) and Maracuyá in Sevilla (Valle). Plantations were selected based on age and harvest records looking for at least three harvests for plantation or exportation quality of the fruit. Figure 7 represents major points of the treatment of fresh pulp in the passifloras. Fruit was collected at ripeness stage 5, the optimum for human consumption [33]. A total of 50−70 units of fruit were collected with the intent of yielding 200 g of lyophilized pulp for subsequent analysis. Fruits were washed in a sodium hypochlorite solution (50 ppm). Fruits were dried with absorbent paper; the pulp was collected without seeds, and then stored at 0 °C until subsequent processing [33].

Fruit pulp was homogenized prior to lyophilization. First, a 40 g aliquot of seedless pulp was homogenized in 100 mL of distilled water using an Ultra-Turrax homogenizer. The homogenization product (juice and pulp) was centrifuged at 1792× *g* for 10 min at room temperature. The supernatants were placed in amber flasks (the remaining pieces of seeds, as precipitate, were discarded) and stored as the working solution at 0 °C until the lyophilization process [34]. Frozen pulp and juice were lyophilized and stored in sealed aluminum bags and kept at 0 °C [33].

### 4.3. Polyphenol Extraction

For characterization of polyphenols, 4 g of each lyophilized fruit were mixed with 30 mL of 70% methanol in water (*v*/*v*), stirred for 20 min at 500 rpm (Dragon Lab MS-H Pro, Beijing, China), sonicated for 15 min (Branson series MH, mod. 3800, St. Louis, MO, USA), and then kept in amber flasks in the dark for at least 24 h, with no stirring [34]. Passiflora extracts were centrifuged (Hermle Z 206 A, Wehingen, Germany) for 10 min at 1372× *g*. Supernatants were utilized for UHPLC-MS analysis.

To prepare polyphenol-rich extracts of pulp for application to cells, ~2 g of each lyophilized *Passiflora* was dissolved in 10 mL of 70% methanol in water (*v*/*v*), vortexed for 3 min at 448× *g*. and sonicated for 20 min. Mixtures were then centrifuged for 15 min at 2800× *g*. The supernatants, with the methanolic polyphenol extracts dried under a nitrogen stream for three hours, were lyophilized (Labconco Freeze Dry, Kansas City, MI, USA) and kept at −20 °C for later UHPLC and Transepithelial Electrical Resistance (TEER) analysis. The precipitate (residue) of these extractions was also saved for later TEER analysis with the purpose of evaluating possible remaining action of non-extracted polyphenols in the lyophilized fruit samples.

### 4.4. UHPLC-MS Analysis of Passion Fruit Polyphenols

Extracts were diluted in methanol:water (0.2% formic acid) in equal proportions, vortexed for 5 min, and sonicated for 5 min preceding chromatographic runs. Peak separation and chromatographic analysis of *Passiflora* extracts were performed in a UHPLC Dionex Ultimate 3000 (Thermo Scientific, Sunnyvale, CA, USA) equipped with a binary pump (HP G3400RS). Polyphenols were separated using a Hypersil GOLD Aq (ThermoScientific, Sunnyvale, CA, USA) 100 × 2.1 mm, 1.9 µm column at 30 °C. Mobile phase A consisted of aqueous ammonium formate (0.2%) and B of acetonitrile with ammonium formate (0.2%). The initial gradient was set at 100% A switching linearly to 100% B over 8 min; then held at 100% B for 4 min before returning to 100% A in 1 min. Total run time was 13 min with 3 min post-runs.

The mass spectrophotometer Exactive Plus Orbitrap (Thermo Scientific, Sunnyvale, CA, USA) was connected to an electrospray ion source (ESI) operated in positive mode with a voltage of 4.5 kV. The spectra were recorded in the range of *m*/*z* 60–900 for full scan MS analysis with nitrogen as the nebulizing gas. The Orbitrap MS detector was calibrated with certified reference solutions Ultramark™ 1621 Mass Spec Std (ABCR GmBH & Co., Karlsruhe, Germany), sodium dodecyl sulphate and hydrated sodium taurocholate (Sigma Aldrich, St. Louis, MO, USA). Compound identification was achieved using full-scan acquisition and ion extraction chromatogram (EIC) mode corresponding to [M + H]^+^ of polyphenols of interest, mass measurement with exactitude, and a precision of Δ_ppm_ < 0.001 using a mixed solution of external phenolic standards, with comparing calibration curves (concentration range from 0.05 to 5.00 µg/mL).

### 4.5. Maintenance of Caco-2 Cells

Human colon adenocarcinoma (Caco-2) cells from ATCC (ATCC^®^ HTB-37™; Manassas, VA, USA) were passaged 20–36 times prior to all experimentation. Cell cultures were maintained in high glucose DMEM and supplemented with 10% (*v*/*v*) FBS, 1% (*v*/*v*) MEM NEAA, and 1% (*v*/*v*) penicillin/streptomycin. Cells were incubated at 37 °C and 5% CO_2_, and subcultured at 80–90% confluence every 3–4 days [35].

### 4.6. Assay for Cell Viability

Confluent cells were detached using trypsin, counted using a Scepter™ 2.0 cell counter (EMD Millipore; Darmstadt, Germany) and seeded at a density of 2 × 10^5^ Caco-2 cells per mL onto polycarbonate membrane Transwell inserts with 0.4 μm pore size [35]. Cells were cultured for 21 days for complete differentiation, and growth media was refreshed every 2–3 days.

### 4.7. Determination of Caco-2 Barrier Dysfunction

The ability of extracts and lyophilized fruit pulp to inhibit inflammation-induced intestinal barrier dysfunction were assessed by continual monitoring of Caco-2 transepitheilial electrical resistance (TEER). All experiments were carried out in a CellZScope+ from NanoAnalytics (Münster, Germany). Cell integrity was evaluated before and during all experiments. Caco-2 cell monolayers registering TEER values of 350–550 ohms∙cm^2^ were utilized for the experimental procedures. The CellZSscope, with inserts with cells in fresh media, was incubated and the instrument was equilibrated for at least 12 h.

Caco-2 cells were treated with an inflammatory cocktail (IC) consisting of inflammatory cytokines IL-1β (25 ng/mL), TNF-α (50 ng/mL), IFN-γ (50 ng/mL), and pro-inflammatory lipopolysaccharide (LPS) (1 mg/mL) for 48 h. The IC was formulated to model intestinal barrier dysfunction induced by chronic inflammation as described previously [24]. Growth media, as a control, of the IC (880 µL) were pipetted into the basolateral compartment of Transwell plates. The lyophilized passiflora pulp products of the methanolic extracts (residues and polyphenols extracts) dissolved in 260 µL of growth media at 2.5, 5.0, and 10 mg/mL were applied to the apical compartments. Cells were incubated at 37 °C and 5% CO_2_ for the duration of the experiment. TEER measurements were registered hourly to 90 h. Results are reported as TEER % versus time.

### 4.8. Statistical Analysis

Data are reported as the mean ± SD or SEM of at least *n* = 3–9 determinations. Statistical analysis was based on initial tests of normality and homogeneity in data and a subsequent Analysis of Variance (ANOVA) followed by Dunnett’s or Tukey HSD tests, where statistical significance was set at *p* < 0.05. GraphPad Prism, version 6.07 (GraphPad, La Jolla, CA, USA) was used for data analysis.

## Figures and Tables

**Figure 1 molecules-24-04614-f001:**
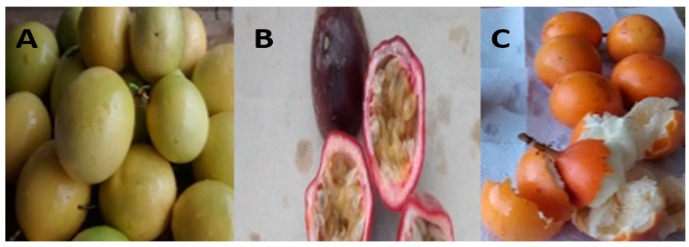
Colombian *Passifloras* (passion fruits). (**A**) Yellow passion fruit. (**B**) Purple passion fruit. (**C**) Sweet granadilla.

**Figure 2 molecules-24-04614-f002:**
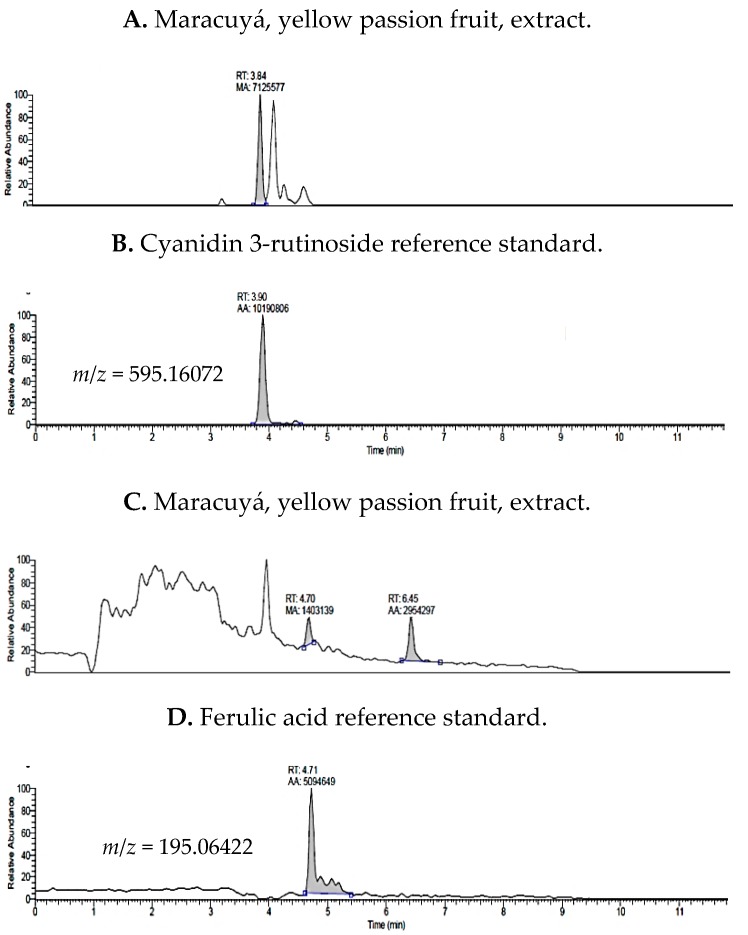
Total Ion Chromatograms (TIC) for yellow passion fruit extracts and comparative standards, cyanidin 3-rutinoside and ferulic acid, evaluated by UHPLC-ESI^+^-Orbitrap-MS.

**Figure 3 molecules-24-04614-f003:**
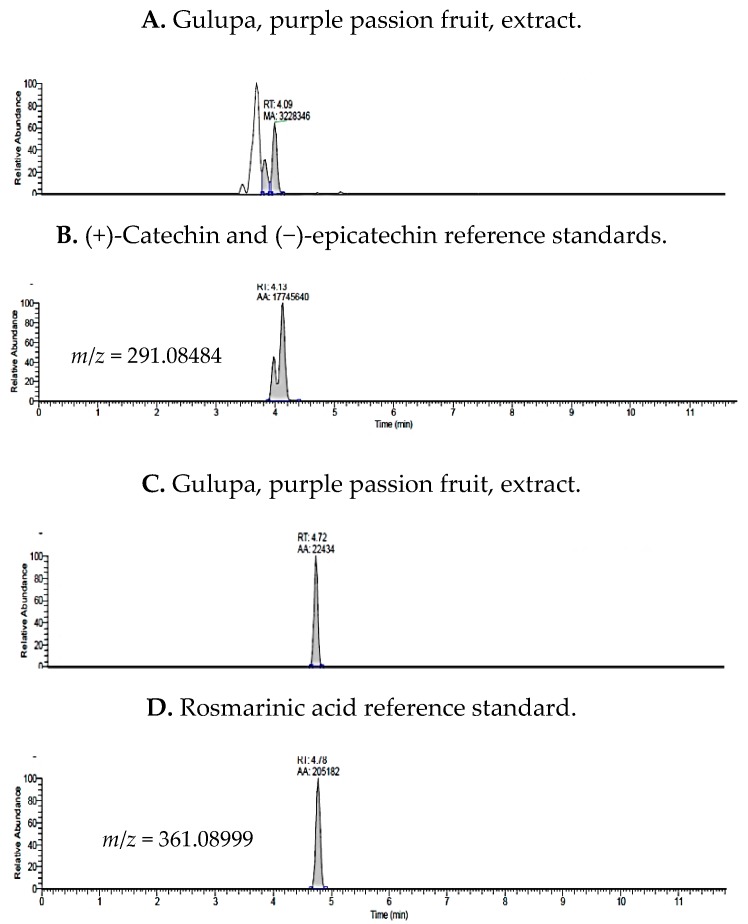
Total Ion Chromatograms (TIC) for purple passion fruit extracts and comparative standards, catechin, epicatechin, and rosmarinic acid.

**Figure 4 molecules-24-04614-f004:**
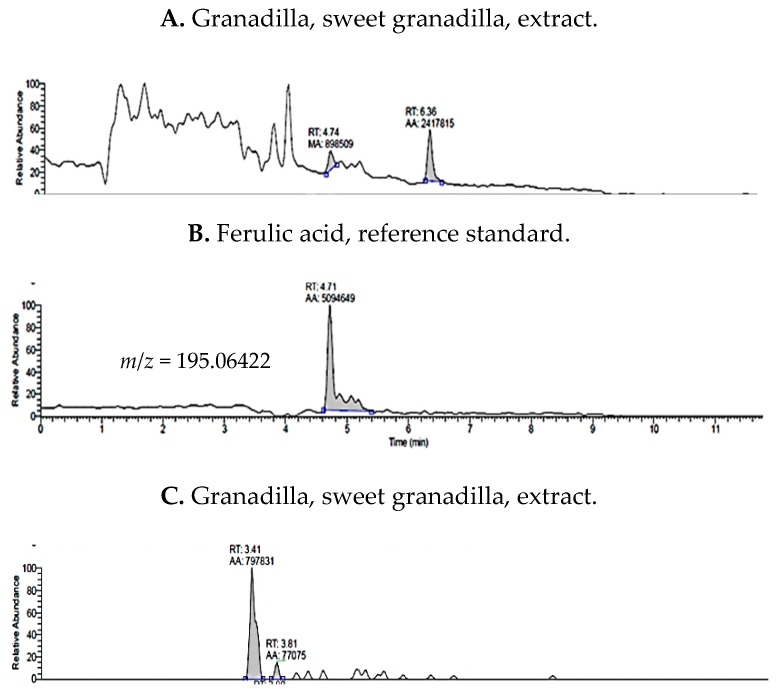
Total Ion Chromatograms (TIC) for sweet granadilla 70% methanol extracts, and comparative standards, ferulic acid, and epigallocatechin.

**Figure 5 molecules-24-04614-f005:**
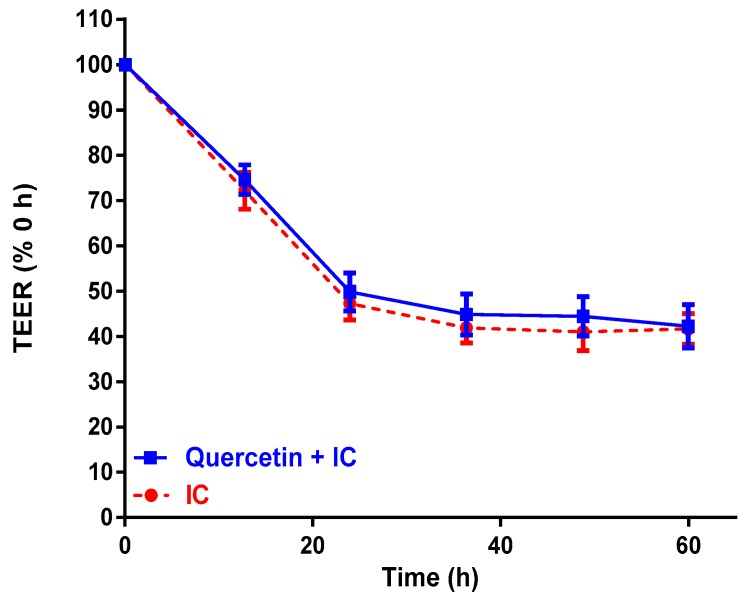
Quercetin does not improve barrier dysfunction of Caco-2 cells treated with an inflammatory cocktail (IC)**.** Differentiated Caco-2 cells were exposed to IC and then monitored for transepithelial electrical resistance (TEER). Quercetin (5 mg/mL, *n* = 5, no significant statistical differences were found between runs for quercetin *p* > 0.05) vs. IC (*n* = 4, no significant statistical differences for this quadruplicate run, *p* > 0.05) with *n* and *p* values for statistical significance (*p* = 0.9065). Data are means ± SDs.

**Figure 6 molecules-24-04614-f006:**
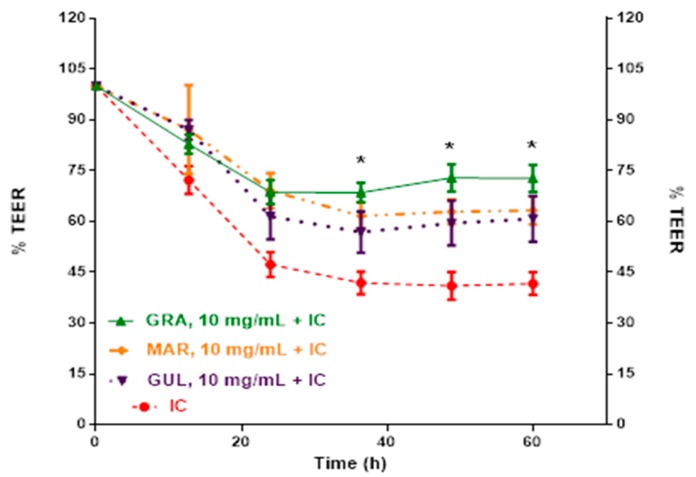
*Passiflora* methanolic extracts inhibit Caco-2 barrier function induced by an inflammatory cocktail (IC). Caco-2 cells treated with methanolic extracts from freeze dried Colombian passifloras Granadilla (GRA) (*n* = 3), Maracuyá (MAR) and Gulupa (GUL) (*n* = 4) and IC. There is a significant statistical difference of 10.0 mg/mL polyphenol extracts with respect to IC (*n* = 8) based on an ANOVA followed by a Dunnett´s test (*p* < 0.0016). All extracts show a significant difference (*) compared to the IC at 36, 48, and 60 h. Data are means ± SDs.

**Figure 7 molecules-24-04614-f007:**
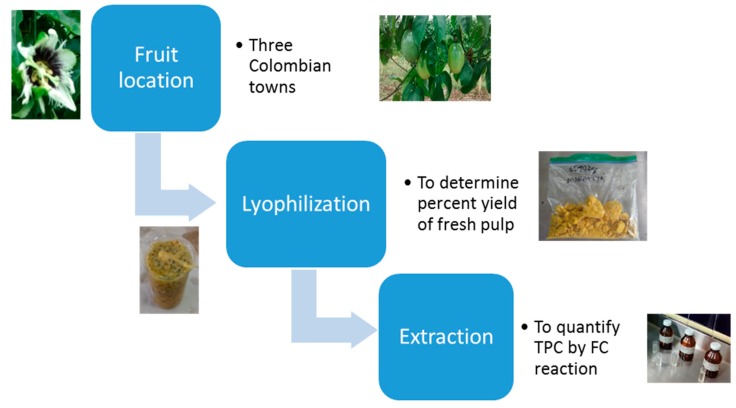
Fresh *Passiflora* pulp preparation for total polyphenol content (TPC) and for a comparative analysis.

**Table 1 molecules-24-04614-t001:** Standard reference compounds, with their retention time and mass spectrometry (MS) responses used for the identification of polyphenols in the three *Passiflora* extracts.

Reference Standards	Retention Time (min)	[M + H]^+^ (*m*/*z*)	*P. edulis* var. Flavicarpa (Maracuyá)	*P. edulis* var. Sims (Gulupa)	*P. ligularis* var. Juss (Granadilla)
theobromine	3.63	181.07109	✓	✓	✓
theophylline	3.78	181.07291	✕	✕	✕
epigallocatechin	3.80	307.07966	✓	✓	✓
cyanidin 3-rutinoside	3.90	595.16072	✓	✓	✓
(+)-catechin	3.97	291.08484	✓	✓	✓
pelargonidin 3-glucoside	4.06	433.11133	✕	✕	✕
caffeine	4.09	195.08672	✓	✓	✓
caffeic acid	4.10	181.04859	✕	✕	✕
(−)-epicatechin	4.13	291.08776	✓	✓	✓
epigallocatechin gallate	4.15	459.08990	✓	✓	✓
vanillic acid	4.16	169.04865	✕	✕	✕
quercetin 3-glucoside	4.45	465.09987	✓	✓	✓
epicatechin gallate	4.49	443.09508	✕	✕	✕
cyanidin	4.55	287.05416	✓	✓	✓
kaempferol 3-glucoside	4.59	449.10558	✕	✕	✕
*p*-coumaric acid	4.63	165.05377	✓	✓	✓
quercetin	4.63	303.04838	✕	✕	✕
ferulic acid	4.71	195.06422	✓	✓	✓
rosmarinic acid	4.78	361.08999	✓	✓	✓
pelargonidin	4.82	271.05928	✓	✓	✓
luteolin	5.24	287.05356	✓	✓	✓
naringenin	5.49	273.07433	✕	✕	✕
apigenin	5.53	271.05874	✓	✓	✓
kaempferol	5.57	287.05644	✕	✕	✕
pinocembrin	6.16	257.07951	✕	✕	✕
carnosic acid	7.94	333.20433	✕	✕	✕
ursolic acid	9.46	457.36631	✓	✓	✓

Presence of the compound in each passiflora extract is represented by the symbol ✓, ✕ indicates compound not detected below 0.02 μg/mL.

**Table 2 molecules-24-04614-t002:** Concentrations of phytochemicals identified in Colombian *Passiflora* 70% methanol extracts (µg/mL) ^a^.

Compounds	*P. edulis* var. Flavicarpa (Maracuyá)	*P. edulis* var. Sims (Gulupa)	*P. ligularis* var. Juss (Granadilla)
cyanidin 3-rutinoside	0.60 ± 0.31	<LOQ ^b^	<LOQ
ferulic acid	0.45 ± 0.30	<LOQ	0.54 ± 0.43
(+)-catechin	0.29 ± 0.02	0.28 ± 0.01	<LOQ
(−)-epicatechin	0.18 ± 0.07	0.22 ± 0.11	0.05 ± 0.01
theobromine	0.09 ± 0.06	<LOQ	<LOQ
quercetin 3-glucoside	0.09 ± 0.06	<LOQ	<LOQ
rosmarinic acid	<LOQ	0.13 ± 0.11	<LOQ
epigallocatechin gallate	<LOQ	<LOQ	0.07 ± 0.04
epigallocatechin	<LOQ	<LOQ	0.09 ± 0.04
caffeine	<LOQ	<LOQ	<LOQ
*p*-coumaric acid	<LOQ	<LOQ	<LOQ
luteolin	<LOQ	<LOQ	<LOQ
apigenin	<LOQ	<LOQ	<LOQ
ursolic acid	<LOQ	<LOQ	<LOQ
cyanidin	<LOQ	<LOQ	<LOQ
pelargonidin	<LOQ	<LOQ	<LOQ

^a^ Data are mean ± SDs with *n* = 3. Where compounds were detected in multiple passifloras varieties, means were not significantly different by ANOVA followed by Tukey’s test (*p >* 0.05). ^b^ LOQ (limit of quantification, based on a calibration curve ranging from 0.05 to 5.00 µg/mL).

**Table 3 molecules-24-04614-t003:** Comparable content of phytochemicals quantified in 70% methanol extracts.

Passifloras/Contents	*P. edulis*	*P. edulis*	*P. ligularis*
	var. Flavicarpa (Maracuyá)	var. Sims (Gulupa)	var. Juss (Granadilla)
Total extract concentration (µg/mL)	1.7 ± 0.82	0.63 ± 0.23	0.75 ± 0.52
Total extract content (µg/g DW)	218.41 ± 0.14	75.22 ± 0.08	96.36 ± 0.13
TPC (mg GAE/g DW)	1.18 ± 0.01	1.62 ± 0.09	1.55 ± 0.00

Data are means ± SDs *(n* = 3). Results for total polyphenol content (TPC) are based on spectrophometrically quantification following Folin-Ciocalteu method (not reported in this work).

**Table 4 molecules-24-04614-t004:** Application of an inflammatory cocktail (IC) reduces barrier function of Caco-2 cells.

Treatment		Time (h)		
	12	24	36	48
Control Media (TEER, % 0 h)	107 ± 6.9 ^a^	106 ± 4.2 ^a^	109 ± 4.0 ^a^	109 ± 2.9 ^a^
IC (TEER, % 0 h)	81 ± 2.6 ^b^	56 ± 1.3 ^b^	51 ± 2.1 ^b^	48 ± 3.5 ^b^

Data are means ± SDs (*n* = 3). Results were tested for ANOVA followed by Dunnett’s test, with *p* < 0.05 considered significant. Columns bearing different letters indicate statistical significance between Control and IC. TEER—transepithelial electrical resistance.

**Table 5 molecules-24-04614-t005:** Treatments (concentrations) from *Passifloras* applied to Caco-2 cells at 48 h.

		TEER %	
Treatments	*P. edulis* var. Flavicarpa (Maracuyá)	*P. edulis* var. Sims (Gulupa)	*P. ligularis* var. Juss (Granadilla)
Residues after extraction	70.65 ± 6.6 ^a^	83.70 ± 6.9 ^b^	71.06 ± 7.1 ^b^
Lyophilized extracts at 2.5 mg/dL	n.d.	n.d.	63.10 ± 3.7 ^a^
Lyophilized extracts at 5.0 mg/dL	62.74 ± 4.0 ^a^	n.d.	63.42 ± 3.6 ^a^
Lyophilized extracts at 10.0 mg/dL	62.88 ± 3.6 ^a^	59.52 ± 6.6 ^a^	72.87 ± 4.0 ^b^

Data points are means ± SD. (n.d. for no data available). Based on ANOVA and post hoc Dunnett’s test (*p* < 0.05) all means (*n* = 3–4) at 48 h are significantly different to the IC (*n* = 8); differences between treatments are represented by different lower-case letters. Residues, after extraction, in Gulupa and Granadilla (*n* = 4) and 10.0 mg/mL extract of Granadilla (*n* = 3) are different with respect to the other treatments.

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
