# Peer review of "Polyphenol Extracts from Three Colombian Passifloras (Passion Fruits) Prevent Inflammation-Induced Barrier Dysfunction of Caco-2 Cells"

_molecules, 2019, doi:10.3390/molecules24244614_

Round 1
Reviewer 1 Report
In the present study the effect of three different species of the Passiflora genus have been characterized and tested for their ability to ameliorate barrier dysfunction induced by a cocktail of pro-inflammatory mediators.
The aim of the manuscript is interesting because botanicals may preserve gut barrier by the action of inflammatory mediators; however, several points lead to have serious concerns about the manuscript, which lacks novelty in its present form.
Concentrations of the extracts used in the study are very high (mg order). Moreover, it is not clear if the extracts per se influence barrier integrity without the presence of inflammatory mediators.
Characterization of the Passiflora extracts has been, at least in part, previously reported in the literature. Although few aspects seem to be novel at this regard, most of them are not.
Studies using in vitro simulated gastrointestinal digestion show that anthocyanins are unstable in the gut, making difficult that that these compounds per se may be able to exert such an effect in the gut. A statement should be provided by authors in the discussion section.
Just measurement of TEER is not enough to state that extracts modify barrier dysfunction. Authors should use other methods (i.e. fluorescence) to demonstrate this effect.
The experiments described in the manuscript are not enough to state that extracts inhibit inflammatory processes. Authors should add additional experiments to demonstrate the anti-inflammatory activity of the extracts. I suggest studying the effect on COX-2, IL-8, IL-6 (proteins and mRNA levels), which are important biomarker of inflammatory processes in the gut.
Which compounds are mostly responsible for the effect observed? The author should discuss also this key point.
Author Response
Dear and respectful Reviewer 1
My co-authors and I thank you for your suggestions that certainly are helping us improve this document.
Comments and Suggestions for Authors, reply by Authors
In the present study the effect of three different species of the Passiflora genus have been characterized and tested for their ability to ameliorate barrier dysfunction induced by a cocktail of pro-inflammatory mediators.
The aim of the manuscript is interesting because botanicals may preserve gut barrier by the action of inflammatory mediators; however, several points lead to have serious concerns about the manuscript, which lacks novelty in its present form.
Concentrations of the extracts used in the study are very high (mg order). Moreover, it is not clear if the extracts per se influence barrier integrity without the presence of inflammatory mediators.
Concentrations were tested from 2.5 to 10 mg/mL due to a low percent yield (13%) considering high humidity percentages in the pulp of all three passifloras. A wider concentration range is expected to be analyzed in the near future.
Non-polar solvents (hexanes and ethyl acetate) were also used in the extraction process (total polyphenol content was very low compared to methanol extracts. TEER experiments were done using hexane and ethyl acetate extracts with fresh media and without the IC. TEER percentages were registered at around 100%.
Characterization of the Passiflora extracts has been, at least in part, previously reported in the literature. Although few aspects seem to be novel at this regard, most of them are not.
Studies using in vitro simulated gastrointestinal digestion show that anthocyanins are unstable in the gut, making difficult that that these compounds per se may be able to exert such an effect in the gut. A statement should be provided by authors in the discussion section.
Although polyphenols exhibited direct effects in the present study, it is possible that instability during digestion or the host and microbial metabolism would change the polyphenol profile in the gut. In example, anthocyanins are extensively catabolized to phenolic acids by gut microbiota [27] Likewise, (-)-epicatechin is catabolized to dihydroxyphenyl-γ-valerolactones in the intestine [28]. Thus, in vivo studies of passion fruits are needed to confirm the activity demonstrated by the present study.
Just measurement of TEER is not enough to state that extracts modify barrier dysfunction. Authors should use other methods (i.e. fluorescence) to demonstrate this effect.
In the planned continuing activities for the present research project, in vivo testing and fluorescence experiments are proposed.
The experiments described in the manuscript are not enough to state that extracts inhibit inflammatory processes. authors should add additional experiments to demonstrate the anti-inflammatory activity of the extracts. I suggest studying the effect on COX-2, IL-8, IL-6 (proteins and mRNA levels), which are important biomarkers of inflammatory processes in the gut.
Which compounds are mostly responsible for the effect observed? The author should discuss also this key point.
More inflammatory-related markers (like the suggested COX-2, IL-8, and IL-6) are to be evaluated in future trials.
Added to Discussion
Considering compound quantification and synergistic effect among them, the suggested compound combination for more responsible of the TEER effects would be for Granadilla ferulic acid (high content) with (-)-epicatechin (low content), for Gulupa ferulic acid (low content) with (+)-catechin and (-)-epicatechin (midlevel content), and for Maracuya ferulic acid (high content) with (+)-catechin and (-)-epicatechin (midlevel content). The compound levels for this comparative aspect is based on 0.05 to 0.15 µg/mL as low contents; 0.16 to 0.30 µg/mL as midlevel contents; and 0.31 to 0.60 µg/mL as high contents. Thus, fruits having ferulic acid with epigallocatechins appeared to be more effective inhibiting IC-induced barrier dysfunction in Caco-2 cells. Prior studies have also demonstrated the ability of FA to inhibit inflammation and protect gastrointestinal cells and TJ [24, 25, 26]. FA in Gulupa, Maracuyá, and Granadilla extracts showed TEER improvement (Table 3) of Caco-2 cells.
We thank you very much for your review and for helping us improve this document.
Reviewer 2 Report
28 treated with the IC. Among the extracts, "P. ligularis" please correct
Caco-2 cells Please "definition" in abstract.
Author Response
Dear and respectful Reviewer 2
My co-authors and I thank you very much for your suggestions that are helping us improve this document.
28 treated with the IC. Among the extracts, "P. ligularis" please correct
Corrected (in Italics)
Caco-2 cells Please "definition" in abstract.
Included (colorectal adenocarcinoma)
We wish a great day.
Reviewer 3 Report
It's a good job, well presented, my suggestions are:
1. From the procedure on Polyphenol extraction, you can make a scheme.
2. You can make the diagrams of Figure 1. of higher quality original chromatograms
3. Was the project authorized by a research committee? You can add it. Thank you
4. You can expand the discussion, I think there is more information to compare your results.
5. Chromatograms schemes, can improve the quality. Thank you
6. in the introduction you can expand it. Thank you
Author Response
Dear and respectful Reviewer 3
My coauthors and I thank you very much for your suggestions that are certainly helping us improve this document.
It's a good job, well presented, my suggestions are:
From the procedure on Polyphenol extraction, you can make a scheme.
Included
You can make the diagrams of Figure 1. of higher quality original chromatograms
Better resolution from Format, Graph Correction
Was the project authorized by a research committee? You can add it. Thank you
All laboratory procedures and fruit sampling were approved by the Bioethical Committee from Universidad de Caldas (Document CBCS-036, Acta No. 012 de 2017).
You can expand the discussion, I think there is more information to compare your results.
Included as per suggestions of Reviewer 1
Chromatograms schemes, can improve the quality. Thank you in the introduction you can expand it. Thank you
Done
Have a wonderful day!
Round 2
Reviewer 1 Report
The authors should add references of studies showing degradation of anthocyanins in the gut, also by simulated gastrointestinal digestion, and provide additional comments.
Author Response
Respectful Reviewer 1
My co-authors and I thank you deeply for your revision and the suggested corrections you kindly offered us. These contributions helped us improve our manuscript.
Have a great day!
Juan Carlos Carmona Hernandez
Lines 257 - 263
[28]. It is important to consider the stability of anthocyanins and other phenolics during digestion. In a human pharmacokinetic study, a significant proportion of 13C cyanidin-3-glucoside was detected as microbial catabolites in the feces [29]. In cell culture media, phenolics have varying stability [30]. Anthocyanins are generally stable during model digestion through the stomach, but less stable during the intestinal phase and form varying phenolic acid products [31]. Grape skins subjected to in vitro digestion retained 40 to 80% of the polyphenols [32].
Reviewer 3 Report
Complies to be published. thanks
Author Response
Dear and Respectful Reviewer 3
My co-authors and I thank you very much for your suggestions that helped us improve this manuscript.
Have a marvelous day!
Juan Carlos Carmona Hernandez